# Altered muscle mitochondrial, inflammatory and trophic markers, and reduced exercise training adaptations in type 1 diabetes

Dean Minnock[1] , Giosuè Annibalini[2], Giacomo Valli[3], Roberta Saltarelli[2], Mauricio Krause[4], Elena Barbieri[2] and Giuseppe De Vito[3]

[1]*School of Public Health, Physiotherapy and Sports Science, University College Dublin, Dublin, Ireland*
[2]*Department of Biomolecular Sciences, University of Urbino Carlo Bo, Urbino, Italy*
[3]*Neuromuscular Physiology Laboratory, Department of Biomedical Sciences, University of Padova, Padova, Italy*
[4]*Department of Physiology, Federal University of Rio Grande do Sul, Porto Alegre, Brazil*

Edited by: Michael Hogan & Bettina Mittendorfer

Linked articles: This article is highlighted in a Perspectives article by Hawke. To read this article, visit https://doi.org/10.1113/JP282800.

The peer review history is available in the Supporting Information section of this article (https://doi.org/10.1113/JP282433#support-information-section).

**Dean Minnock** was awarded the Irish Research Council (IRC) Employment-Based PhD fellowship in 2016. As an IRC fellow, he conducted his PhD research at the University College Dublin (UCD), as a member of Professor Giuseppe De Vito's research group. His research work is focused at the intersection of integrative molecular and exercise physiology, examining physical interventions on metabolism and molecular regulation of skeletal muscle to reverse or regress diabetic complications. **Giosuè Annibalini** is a researcher at the University of Urbino Carlo Bo, Italy. His research focuses on the molecular mechanism underlying the health benefits of exercise, with particular interest in the exercise-induced regulation of inflammation and growth-related signalling pathways.

 

D. Minnock and G. Annibalini are co-first authors.

**Abstract**  Growing evidence of impaired skeletal muscle health in people with type 1 diabetes points toward the presence of a mild myopathy in this population. However, this myopathic condition is not yet well characterised and often overlooked, even though it might affect the whole-body glucose homeostasis and the development of comorbidities. This study aimed to compare skeletal muscle adaptations and changes in glycaemic control after 12 weeks of combined resistance and aerobic (COMB) training between people with type 1 diabetes and healthy controls, and to determine whether the impaired muscle health in type 1 diabetes can affect the exercise-induced adaptations. The COMB training intervention increased aerobic capacity and muscle strength in both healthy and type 1 diabetes sedentary participants, although these improvements were higher in the control group. Better glucose control, reduced glycaemic fluctuations and fewer hypoglycaemic events were recorded at post- compared to pre-intervention in type 1 diabetes. Analysis of muscle biopsies showed an alteration of muscle markers of mitochondrial functions, inflammation, ageing and growth/atrophy compared to the control group. These muscular molecular differences were only partially modified by the COMB training and might explain the reduced exercise adaptation observed in type 1 diabetes. In brief, type 1 diabetes impairs many aspects of skeletal muscle health and might affect the exercise-induced adaptations. Defining the magnitude of diabetic myopathy and the effect of exercise, including longer duration of the intervention, will drive the development of strategies to maximise muscle health in the type 1 diabetes population.

(Received 29 September 2021; accepted after revision 21 December 2021; first published online 7 January 2022)

**Corresponding author** G. Valli: Neuromuscular Physiology Laboratory, Department of Biomedical Sciences, University of Padova, via F. Marzolo 3, 35131 Padova, Italy.    Email: giacomo.valli@phd.unipd.it

**Abstract figure legend** The effect of type 1 diabetes on functional and molecular markers of skeletal muscle health in response to a training period and compared to healthy people. Ten people with type 1 diabetes and 10 healthy control subjects participated in a 12-week, high-intensity, combined exercise training programme. Muscle biopsies were collected before and after the training. At baseline, participants with type 1 diabetes exhibited a reduced level of mitochondrial DNA, proteins related to the oxidative phosphorylation (OXPHOS) system and reduced telomere length compared to control. These changes were detectable despite the two groups being comparable for body composition and functional measurements. The training programme improved glycaemic control but people with type 1 diabetes had a reduced adaptation in muscle force and aerobic capacity compared to control. The training period only partially improved the diabetes-induced alterations of skeletal muscle health. Created with BioRender.com.

### Key points

- Type 1 diabetes negatively affects skeletal muscle health; however, the effect of structured exercise training on markers of mitochondrial function, inflammation and regeneration is not known.
- Even though participants with type 1 diabetes and healthy control were comparable for cardio-respiratory fitness ($\dot{V}_{O_2max}$) and muscle strength at baseline, molecular markers related to muscle health were decreased in type 1 diabetes.
- After training, both groups increased $\dot{V}_{O_2max}$ and muscle strength; however, a larger improvement was achieved by the control group.
- The training intervention decreased glucose fluctuations and occurrence of hypoglycaemic events in type 1 diabetes, while signs of mild myopathy found in the muscle of participants with type 1 diabetes only partially improved after training
- Improving muscle health by specific exercise protocols is of considerable clinical interest in therapeutic strategies for improving type 1 diabetes management and preventing or delaying long-term complications.

## Introduction

Skeletal muscle plays a major role in the regulation of whole-body energy metabolism and glucose homeo-stasis. As a consequence, reduced muscle health can be detrimental for metabolic control (Periasamy *et al.* 2017) and heavily influence the occurrence of long-term

complications (Daneman, 2006). Studies in rodents suggest that type 1 diabetes exposes muscle to damage (D'Souza *et al.* 2016) and delayed regeneration (Krause *et al.* 2013). In humans, type 1 diabetes is associated with muscle quantitative and functional pejorative alterations such as reduced myofibre diameter, strength and increased fatigability (Monaco *et al.* 2017). These often-overlooked complications contribute to the definition of diabetic myopathy, which might be a factor in the development of glucose dysregulation and diabetic complications (Janssen *et al.* 2004).

Despite growing information related to loss of muscle health in people living with type 1 diabetes, the specific triggering events remain largely unknown. Recently, alterations in mitochondrial ultrastructure, bioenergetics and mitochondrial area density were identified as the most evident impairment in the muscle of humans with type 1 diabetes (Monaco *et al.* 2018, 2021).

An important molecular factor driving mitochondrial biogenesis is peroxisome proliferator-activated receptor γ coactivator 1α (PGC-1α), which controls the expression of genes involved in oxidative phosphorylation (OXPHOS) including the cytochrome *c* oxidase subunits (e.g. the nuclear-encoded *COX5B* and the mitochondrially encoded *MT-CO2*) (Mootha *et al.* 2003; Sivitz & Yorek, 2010). The activation of this pathway is enhanced by activation of AMP-activated protein kinase (AMPK), a metabolic sensor enzyme responding to states of ATP depletion as in reduced caloric intake and exercise (Cantó & Auwerx, 2009).

Altogether, these findings recapitulate a condition of accelerated skeletal muscle ageing (Dial *et al.* 2021) associated with the decline in mitochondrial function in type 1 diabetes (Monaco *et al.* 2019; Heyman *et al.* 2020).

Regular physical activity is the primary approach for the maintenance and promotion of skeletal muscle health and function also in type 1 diabetes, but depending on type, duration and intensity, exercise can heavily affect glycaemic control (Riddell *et al.* 2017; Valli *et al.* 2021).

Recently, we evaluated the acute effects of high-intensity aerobic (AER), resistance (RES) and combined RES + AER (COMB) exercise protocols on glucose control and skeletal muscle molecular responses of participants with type 1 diabetes (Minnock *et al.* 2020). Interestingly, COMB was the most effective in reducing interstitial glucose (IG) variability and simultaneously activated muscle signalling pathways involved in substrate metabolism and anabolic adaptations. These findings suggest a possibility for ameliorating glucose homeostasis and muscle health in people with type 1 diabetes by COMB exercise training.

Therefore, this study was conducted to investigate the effect of 12 weeks of high-intensity COMB exercise training comparing people with type 1 diabetes to healthy controls. The main parameters of interest were gene expression and signalling pathways related to metabolism, mitochondrial content, inflammation and muscular growth that are known to be associated with exercise-induced functional adaptations and skeletal muscle health. Moreover, we measured IG variability and the occurrence of hypoglycaemic events.

We wanted to verify whether the positive adjustments seen in acute exercise could translate into chronic adaptations and whether the skeletal muscle condition at baseline, or in response to exercise, differed between type 1 diabetes and healthy participants.

## Methods

### Ethical approval

This study was approved by the Human Research Ethics Committee of University College Dublin (LS-17-113-Minnock-DeVito) and conformed to the standards set by the *Declaration of Helsinki*. Comprehensive details of the study protocol were explained orally and in writing prior to participants providing their written informed consent to partake in the intervention.

### Participant information

Twenty participants, 10 type 1 diabetes (4 M/6 F; 31.6 ± 3.7 years) without comorbidities, and ten healthy participants (4 M/6 F; 28.4 ± 5.9 years) were enrolled in the study. Sample size was based on power analysis on flash glucose monitoring (FGM) data from our previous study (Minnock *et al.* 2020). Inclusion criteria were age between 18 and 45 years, type 1 diabetes diagnosed for at least 12 months, BMI <30 kg/m$^2$, non-smokers, absence of comorbidities, complications and any other condition preventing participation in the intervention. Participants were not habitual exercisers (less than 2 h of physical activity per week (Active Australia Survey) and mean metabolic equivalent of task (MET) expenditure <500 MET weekly (GPAQ scores)). The type 1 diabetes group consisted of nine multiple daily injection users and one insulin pump user with an HbA1c of 7.92 ± 3.43% (63 ± 14 mmol/mol) and diabetes duration of 17.8 ± 9.2 years. All female participants were using oral hormonal contraceptives.

### Exercise training intervention

The 12-week training intervention consisted of three weekly 40 min, high-intensity COMB exercise sessions divided into 20 min RES and 20 min AER exercises as previously described (Minnock *et al.* 2020). Every session was supervised by experienced trainers. During each session, participants were fitted with a heart

rate monitor (Polar H7, Kempele, Finland). The Karvonen formula was applied to establish the target 80% heart rate reserve (HRR) zone for the AER session (Minnock *et al.* 2020).

Briefly, each session started with a warm-up period followed by high-intensity RES and AER exercises. During RES, participants performed six different exercise techniques combining free-weights and resistance machines (goblet squat (GOB), bicep curl (BC), leg extension (LE), seated cable row (SCR), squat press (SQUAT), and triceps extension (TE)) at 80% of their one-repetition maximum (1RM) (corresponding to 8–10 repetitions). During AER, participants pedalled on a cycle ergometer (Excalibur, Lode BV, Groningen, The Netherlands) at 80 rpm adjusting the resistance until the desired 80% HRR zone was reached. Intervention ceased once 40 min of training had been completed. All participants were advised and encouraged to consume water *ad libitum* throughout the exercise session to mini-mise cardiac drift. Session rate of perceived exertion (RPE) for each participant was recorded using a CR-10 scale at the end of each session (Borg, 1998). Correct technique for RES using lower loads (60% 1RM) was taught in week 1. Training load was closely monitored and adjusted when necessary to maintain the pre-scribed number of repetitions. Participants recorded their diet at pre-intervention to be replicated in the post-intervention collection week to ensure no changes occurred to their nutritional habits. The type 1 diabetes group adhered to their normal insulin therapy and female participants continued their contraception therapy. No other medication was used in either group.

## Body composition, maximum oxygen consumption and muscle strength

Body composition was assessed by dual-energy X-ray absorptiometry (DEXA) (Lunar iDXA, GE Healthcare, London, UK). Immediately following the DEXA scan, resting heart rate was recorded (Omron M2, Kyoto, Japan). To assess the maximum oxygen consumption ($\dot{V}_{O_2max}$) and maximum heart rate, participants performed an incremental maximal cycling test to volitional exhaustion using an open circuit metabolimeter (Quarkb2 Cosmed, Rome, Italy) on a cycle ergometer (Excalibur Sport). At Pre-intervention, 1RM for each RES exercise was established after a familiarisation session.

## Interstitial glucose monitoring

IG was monitored using FGM devices (Abbott Freestyle Libre, Alameda, CA, USA). FGM patches were provided to each participant at the first and the last week of the 12-week training programme. IG was recorded every 15 min and the data were downloaded to provide IG data for week-long periods. IG variability was evaluated using the following four indices: mean amplitude of glycaemic excursions (MAGE), IG variance (VAR), IG coefficient of variation (CV) and IG standard deviation (SD) (Zaccardi *et al.* 2008; Hill *et al.* 2011). MAGE, VAR, CV, and SD analysis was performed on the IG data collected during the entire week-long periods. In particular, the MAGE calculation was based on the differences between consecutive points, considering those that were more than 1SD higher than the previous point (Hill *et al.* 2011).

## Muscle sampling

Muscle biopsies were collected at pre- and post-intervention using a 14-gauge semi-automatic spring-loaded biopsy system with a compatible coaxial introducer needle (Medax Srl Unipersonale; San Possidonio, Italy). Participants refrained from exercising for at least 48 h before muscle sampling. The biopsies were taken at rest from both left and right *vastus lateralis* muscle, immediately frozen in liquid nitrogen and stored at −80°C (Minnock *et al.* 2020).

## Muscle nucleic acid and protein extraction and analysis

Without thawing, all muscle tissues were weighed (≤30 mg), placed directly into the QIAzol Lysis Buffer (Qiagen, Milan, Italy), and ruptured using a Polytron homogeniser (Kinematica AG, Switzerland). Total RNA was extracted and analysed by real-time PCR as pre-viously described (Minnock et al. 2020). Total DNA was isolated from the organic phase of the QIAzol Lysis Buffer according to the manufacturer's instructions and purified using the E.Z.N.A. Tissue DNA Kit (OMEGA Bio-Tek, Norcross, GA, USA). The amount of DNA was assessed with the SpectraMax QuickDrop Micro-Volume UV-Vis Spectrophotometer (Molecular Devices, San Jose, CA, USA). Subsequently, quantitative real-time PCR performed with 100 ng of DNA and 500 nM of each primer in an Applied Biosystems StepOnePlus™ Real-Time PCR System using PowerUp SYBR Green Master Mix (Thermo Fisher Scientific, Monza, Italy). Nuclear (nDNA) and mitochondrial DNA (mtDNA) content from muscle samples were measured by real-time PCR using specific primers for *36B4* (nuclear) and mitochondrially encoded *MT-CO1* (Guescini *et al.* 2007) under the following real-time PCR conditions: 50°C for 2 min, 95°C for 2 min followed by 40 cycles of two-steps at 95°C for 15 s and 60°C for 60 s. Telomere length was measured by real-time PCR using the ratio of telomeric repeats copy number (T) to a single copy (36B4) reference gene (S), as described by Cawthon (2002). The telomere and single-copy gene (*36B4*) were analysed on the same plate to reduce interassay variability. The real-time PCR

**Table 1. Primers used in real-time RT-PCR quantification**

| Gene | Primer forward (5′–3′) | Primer reverse (5′–3′) |
|---|---|---|
| *IL-6* | GGTACATCCTCGACGGCATCT | GTGCCTCTTTGCTGCTTTCAC |
| *TNF-α* | GACAAGCCTGTAGCCCATGT | TCTCAGCTCCACGCCATT |
| *MCP-1* | CGCCTCCAGCATGAAAGTC | GTGACTGGGGCATTGATTG |
| *IL-8* | GCAGAGGGTTGTGGAGAAGT | GCTTGAAGTTTCACTGGCATC |
| *IGF-1Ea* | GACATGCCCAAGACCCAGAAGGA | CGGTGGCATGTCACTCTTCACTC |
| *IGF-1Eb* | CTACCAACAAGAACACGAAGT | CTACCAACAAGAACACGAAGT |
| *IGF-1Ec* | GCCCCCATCTACCAACAAGAACAC | TCCCTCTACTTGCGTTCTTCAAA |
| *PGC-1α* | ATGGAGTGACATCGAGTGTGCT | GAGTCCACCCAGAAAGCTGT |
| *MT-CO1* | CTGCTATAGTGGAGGCCGGA | GGGTGGGAGTAGTTCCCTGC |
| *MT-CO2* | AGATGCAATTCCCGGACGT | CATGAAACTGTGGTTTGCTCC |
| *COX5B* | ACAATGTACTGGCCCCAAAGG | TTGTAATGGGCTCCACAGC |
| *Telomere length* | GGTTTTTGAGGGTGAGGGTGAGGGTGAGGGTGAGGGT | TCCCGACTATCCCTATCCCTATCCCTATCCCTATCCCTA |
| *MuRF1* | CCTGAGAGCCATTGACTTTGG | CTTCCCTTCTGTGGACTCTTCCT |
| *Atrogin-1* | GCAGCTGAACAACATTCAGATCAC | CAGCCTCTGCATGATGTTCAGT |
| *Myostatin* | CCAGGAGAAGATGGGCTGAA | CAAGACCAAAATCCCTTCTGGAT |
| *B2M* | GATCGAGACATGTAAGCAGC | CAAACATGGAGACAGCACTC |
| *GAPDH* | AAATCAAGTGGGGCGATGCT | TGCTGATGATCTTGAGGCTG |
| *36B4* | CAGCAAGTGGGAAGGTGTAATCC | CCCATTCTATCATCAACGGGTACAA |

*36B4*, ribosomal protein lateral stalk subunit P0; *Atrogin-1*, F-box protein 32; *B2M*, beta-2-microglobulin; *COX5*, cytochrome *c* oxidase subunit 5A; *GAPDH*, gliceraldeyde-3-phosphate dehydrogenase; *IGF-1*, insulin-like growth factor-1; *IL-6*, interleukin-6; *IL-8*, interleukin-8; *MCP-1*, monocyte chemotactic protein 1; *MT-CO1*, cytochrome *c* oxidase subunit I; *MT-CO2*, cytochrome *c* oxidase sub-unit II; *MuRF1*, tripartite motif containing 63; *PGC-1*, peroxisome proliferator-activated receptor $\gamma$ coactivator 1-$\alpha$; *TNF-α*, tumour necrosis factor-$\alpha$.

conditions for telomere length quantification were 50°C for 2 min, 95°C for 2 min followed by 40 cycles of three-steps at 95°C for 15 s, 54°C for 15 s, and 72°C for 20 s. The sequence of primers used in real-time PCR is listed in Table 1.

Total protein extracts were obtained from the organic phase following the QIAzol protocol, solubilised in lysis buffer containing 20 mM HEPES (pH 7.9), 25% v/v glycerol, 0.42 M NaCl, 0.2 mM EDTA, 1.5 mM MgCl$_2$, 0.5% v/v Nonidet P-40, 1 mM dithiothreitol, 1 mM NaF, 1 mM Na$_3$VO$_4$, and 1 × complete protease inhibitor cocktail (Roche Diagnostics GmbH, Mannheim, Germany) and processed for western blot analysis as previously reported (De Santi *et al.* 2016). Briefly, 40 $\mu$g of total proteins was electrophoresed with 10% SDS-PAGE, and then transferred to nitrocellulose or polyvinylidene difluoride membranes (Bio-Rad Laboratories, Hercules, CA, USA) for immunoblotting. The primary antibodies from Cell Signaling Technology (Danvers, MA, USA) were used diluted (1:2000) in Tris-buffered saline–Tween 20 (TBS-T) containing 5% BSA: phospho-AMPKα (Thr172, 40H9, cat. no. 2535), AMPKα (cat. no. 2532), phospho-p38 mitogen-activated protein kinase (MAPK) (Thr180/Tyr182, cat. no. 9211), p38 MAPK (cat. no. 9212), inhibitor of nuclear factor $\kappa$B kinase (IKK)-$\beta$ (D30C6, cat. no. 8943), phospho-Akt (Ser473, cat. no. 9271), Akt (cat. no. 9272), phospho-p44/42 MAPK (extracellular signal-regulated kinase (ERK)-1/2) (Thr202/Tyr204,

cat. no. 9101), p44/42 MAPK (ERK1/2, cat. no. 9102), phospho-eEF2 (Thr56, cat. no. 2331) and eEF2 (cat. no. 2332). In addition, human total oxidative phosphorylation (OXPHOS) cocktail from Abcam (Cambridge, MA, USA, cat. no. ab110411) was used diluted 1:1000 to detect individual complexes of the electron transport chain (I, II, III, IV, V).

### Statistical analysis

Descriptive data are reported as means $\pm$ SD. Parameters of interest were checked for normal distribution using the Shapiro–Wilks test and parametric statistical analysis was employed to determine significant differences. Results for $\dot{V}_{O_2max}$ and *TNF-α* mRNA expression were not normally distributed and therefore log-transformed and reanalysed. To verify changes between groups during the exercise intervention two-way mixed design multivariate analysis of variance (MANOVA) with Bonferroni *post hoc* test was used. Time was within-participants, while control and type 1 diabetes were two level between-group factors. To verify changes within the type 1 diabetes group (e.g. number of hypoglycaemic events), a two-tailed paired Student's *t*-test was used. Due to the small sample size and the sex distribution (4 M/6 F) it was not possible to make any satisfactory investigation of between sex differences. Data were analysed using SPSS Statistics

**Table 2. Baseline to follow-up change in body composition, muscle strength and cardiorespiratory fitness**

| | Con Pre | Con Post | T1D Pre | T1D Post | Between (Group) difference $P$-value | Within (Time) difference $P$-value | Time x Group interaction $P$-value |
|---|---|---|---|---|---|---|---|
| Body mass (kg) | 69.69 ± 12.00 | 69.38 ± 11.51 | 77.15 ± 11.22 | 77.96 ± 12.38 | 0.143 | 0.691 | 0.378 |
| Body mass index (kg/m$^2$) | 23.95 ± 2.85 | 23.85 ± 2.63 | 26.51 ± 3.43 | 26.81 ± 4.17 | 0.077 | 0.651 | 0.377 |
| Body fat (%) | 29.39 ± 6.67 | 28.70 ± 6.57 | 30.12 ± 9.40 | 29.24 ± 9.59 | 0.863 | 0.100 | 0.836 |
| Lean body mass (kg) | 47.00 ± 9.25 | 46.96 ± 9.20 | 53.55 ± 14.31 | 52.45 ± 11.24 | 0.250 | 0.718 | 0.734 |
| $\dot{V}_{O_2max}$ (ml/kg/min) | 32.76 ± 5.20 | 41.72 ± 3.71*** | 31.55 ± 9.55 | 34.98 ± 9.85***,† | 0.216 | <0.0001 | <0.0001 |
| GOB (kg) | 49.30 ± 15.76 | 58.30 ± 18.17*** | 42.60 ± 21.38 | 47.90 ± 22.81*** | 0.342 | <0.0001 | 0.066 |
| BC (kg) | 9.95 ± 3.45 | 12.50 ± 4.65** | 10.25 ± 4.04 | 11.30 ± 4.42** | 0.808 | <0.0001 | 0.067 |
| LE (kg) | 56.4 ± 23.67 | 67.4 ± 21.4*** | 69.60 ± 19.21 | 73.70 ± 18.65* | 0.307 | <0.0001 | 0.001 |
| SCR (kg) | 51.70 ± 17.43 | 60.30 ± 16.46*** | 46.30 ± 9.99 | 54.20 ± 11.49*** | 0.373 | <0.0001 | 0.687 |
| SQUAT (kg) | 108.6 ± 14.39 | 134.1 ± 10.71*** | 107.60 ± 18.08 | 124.60 ± 17.90*** | 0.446 | <0.0001 | 0.023 |
| TE (kg) | 15.20 ± 4.05 | 16.60 ± 3.78*** | 14.10 ± 3.67 | 15.70 ± 3.33*** | 0.551 | <0.0001 | 0.667 |

Values are presented as means ± SD. Statistical differences were determined by two-way repeated MANOVA. Significantly different compared to Pre:
*$P < 0.05$, **$P < 0.01$, ***$P < 0.001$; †significantly different compared to Con at the same time point: †$P < 0.05$. BC, bicep curl; Con, control group; GOB, goblet squat; LE, leg extension; SCR, seated cable row; SQUAT, squat press; T1D, participants with type 1 diabetes; TE, triceps extension.

Version 20.0 (IBM Corp., Armonk, NY, USA). For all analyses, statistical significance was accepted at $P < 0.05$.

## Results

### Body composition, cardiorespiratory fitness, and muscle strength

All participants completed the study without complications or injuries. Table 2 shows data and $P$-values for within- and between-arm comparisons and interaction terms for body composition, cardiorespiratory fitness and muscle strength. There were no differences between the two groups for all the tested variables at baseline. Body composition did not change after the exercise training in both groups. There was a significant interaction between group and time for $\dot{V}_{O_2max}$, LE and SQUAT, while GOB ($P = 0.066$) and BC ($P = 0.067$) approached the borderline of significance. Both arms showed a significant increase in $\dot{V}_{O_2max}$ and 1RM, but a larger improvement was achieved by the control group. *Post hoc* comparisons between groups indicated that the control group had a higher $\dot{V}_{O_2max}$ level than type 1 diabetes at post-intervention.

To exclude that the cause of minor adaptations in type 1 diabetes was non-compliance with exercise prescription, exercise intensity was validated by the measure of session RPE. Session RPE was similar (not statistically different) across all the weeks of intervention except at weeks 10 and 12 where it was found to be higher for type 1 diabetes (not shown).

Carbohydrate intake was higher in type 1 diabetes compared to the control group ($P = 0.001$) but did not differ at pre- and post-intervention week ($P = 0.675$) (control: 1356.00 ± 331.28 g Pre, 1382.50 ± 283.31 g

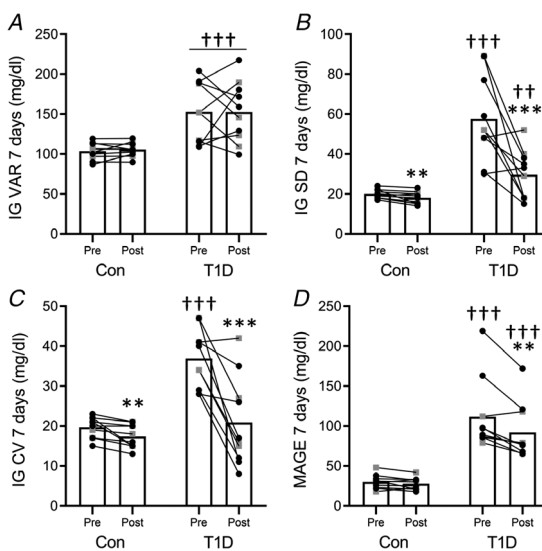

**Figure 1. Glycaemic data over the weeklong period pre- and post-training**
Interstitial glucose (IG) profile was used to calculate interstitial glucose variance (IG VAR) (*A*) standard deviation (IG SD) (*B*), coefficient of variation (CV) (*C*), and the mean amplitude of glycaemic excursions (MAGE) (*D*). All the panels show the pre- and post-training measurements for the control group (Con) and for participants with type 1 diabetes (T1D) (female: black circles; male: grey squares). *$P < 0.05$, **$P < 0.01$, ***$P < 0.001$ compared to Pre; ††$P < 0.01$, †††$P < 0.001$, compared to control.

Post; type 1 diabetes: 2014.30 ± 418.09 g Pre, 2013.90 ± 394.56 g Post). Insulin intake in type 1 diabetes did not change from pre- to post-training (151.90 ± 29.47 units Pre, 151.00 ± 29.44 units Post; $P = 0.430$).

## Glucose variability

There was no statistically significant interaction between group and time on IG VAR (Fig. 1*A*). The main effect of the group showed that IG VAR was higher in type 1 diabetes compared to the control group. There was a significant interaction between group and time for IG SD, CV, and MAGE (Fig. 1*B–D*). In both arms, IG SD (Fig. 1*B*)

and IG CV (Fig. 1*C*) decreased after the training period; however, the reduction was higher in type 1 diabetes compared to the control group. Conversely, MAGE decreased only in type 1 diabetes at post-intervention (Fig. 1*D*). *Post hoc* comparisons between groups indicated higher IG SD, IG CV and MAGE in type 1 diabetes at pre-intervention, while only IG SD and MAGE differed between groups at post-intervention.

## Hypoglycaemic events

During the pre-intervention week, all participants with diabetes experienced hypoglycaemic events while

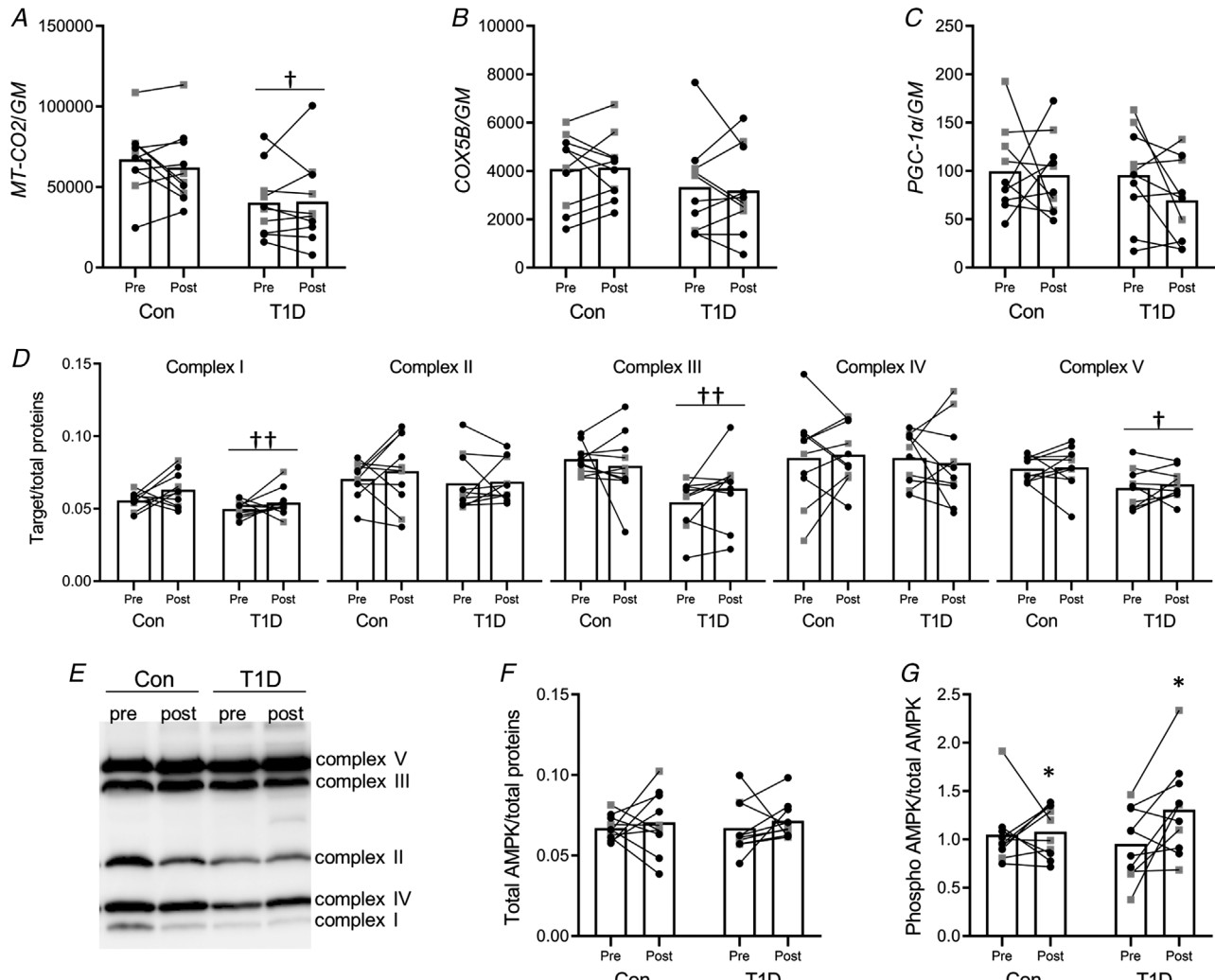

**Figure 2. Markers of mitochondrial function**
*A–C*, muscle mRNA expression levels of *MT-CO2* (*A*), *COX5B* (*B*) and *PGC-1α* (*C*) measured by real-time PCR. The geometric mean (GM) of *GAPDH* and *B2M* mRNA levels was used as the internal reference. *D* and *E*, representative western blot image (*E*) and plot (*D*) of the densitometric analyses of complexes I, II, III, IV and V of the OXPHOS system normalised on total proteins. *F* and *G*, densitometric analyses of total (*F*) and phosphorylated (*G*) AMPK respectively normalised to total proteins and total AMPK. All the panels show the pre- and post-training measurements for the control group (Con) and for participants with type 1 diabetes (T1D) (female: black circles; male: grey squares). *$P < 0.05$, compared to Pre; †$P < 0.05$, ††$P < 0.01$, compared to control.

seven participants experienced hypoglycaemic events at post-intervention. The total number of reported events significantly diminished between pre- ($10.60 \pm 4.92$) and post-intervention weeks ($2.10 \pm 1.92$). The weeklong average duration of hypoglycaemic events measured by flash glucose monitoring did not change between pre- ($378.67 \pm 383.28$ min) and post-intervention ($281.26 \pm 530.87$ min).

### Markers of mitochondrial function and ageing

There was no significant interaction between group and time on mitochondria-related gene expression (*MT-CO2*, *COX5B* and *PGC-1a*) (Fig. 2*A*–*C*), OXPHOS protein complex level (Fig. 2*D*) and total AMPK level or AMPK phosphorylation (Fig. 2*F* and *G*). However, a main group effect showed that *MT-CO2* mRNA level (Fig. 2*A*) and relative level of complex I, complex III and complex V of the OXPHOS system were lower in type 1 diabetes compared to control (Fig. 2*D*). The main effect of time showed increased AMPK phosphorylation after training (Fig. 2*G*).

There was no significant interaction between group and time on mtDNA content (Fig. 3*A*) and telomere length (Fig. 3*B*). The main effect of the group showed that mtDNA content and muscle telomere length were lower in type 1 diabetes than control.

### Markers of inflammation

There was no significant interaction between group and time on the inflammation-related gene (*TNF-a*, *IL-6*, *IL-8* and *MCP-1*) expression (Fig. 4*A*–*D*). However, the main effect of group showed that *TNF-a* mRNA level was higher in type 1 diabetes than control (Fig. 4*A*). The main effect of time showed an increase of *MPC-1* mRNA level after the training programme. There was a significant interaction between group and time on p38 MAPK phosphorylation (Fig. 4*F*). The p38 MAPK phosphorylation level increased after the exercise training only in the control group (Fig. 4*F*). *Post hoc* comparisons between groups indicated that baseline p38 MAPK phosphorylation level was higher in type 1 diabetes than control. Total p38 MAPK (Fig. 4*E*) and IKK$\beta$ (Fig. 4*G*) levels did not change.

### Markers of muscle growth, remodelling and atrophy

There was a significant interaction between group and time on *IGF-1* isoform expression (Fig. 5*A*–*C*), showing that all *IGF-1* mRNA variants increased after training only in type 1 diabetes. *Post hoc* comparisons between groups showed that type 1 diabetes had higher *IGF-1Ea* and *IGF-1Ec* mRNA levels than control at post-intervention (Fig. 5*A* and *C*). There was no significant interaction between group and time on total and phosphorylated eEF2, Akt and ERK1/2 protein levels. The main effect of the group showed that eEF2 phosphorylation (i.e. inactivation) was higher in type 1 diabetes than control (Fig. 5*I*). The main effect of time showed an increase of total Akt (Fig. 5*D*) and ERK1/2 phosphorylation level (Fig. 5*G*) after training. Phospho-Akt (Fig. 5*E*), total ERK1/2 (Fig. 5*F*) and total eEF2 (Fig. 5*H*) level did not change. There was a significant interaction between group and time on the *MuRF1* mRNA level, i.e. *MuRF1* mRNA expression decreased only in type 1 diabetes after training (Fig. 6*A*). *Post hoc* comparisons between groups indicated that baseline *MuRF1* mRNA level was higher in type 1 diabetes. *Atrogin-1* (Fig. 6*B*) and *Myostatin* (Fig. 6*C*) mRNA expression did not change.

### Discussion

This study demonstrated that 12 weeks of high-intensity COMB RES and AER exercise improved glycaemic control and reduced the occurrence of hypoglycaemic events in participants with type 1 diabetes. Furthermore, COMB training enhanced muscle strength and cardiorespiratory fitness in both type 1 diabetes and control groups. However, the exercise adaptations, although significant, were less pronounced in type 1 diabetes. It is noteworthy that participants with type 1 diabetes presented baseline alterations in muscle markers of mitochondrial function, inflammation and muscle growth/atrophy which were only partially improved by the training intervention.

The reduced glucose variability observed after COMB training in type 1 diabetes agrees with our previous findings in which acute COMB exercise was effective in decreasing the post-exercise glucose fluctuations

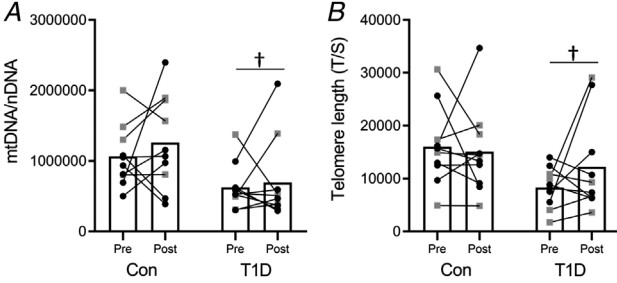

**Figure 3. Markers of ageing**
Muscle content of mitochondrial DNA (mtDNA) (*A*) and telomere length (*B*) measured by real-time PCR. The ratio of mitochondrial *MT-CO1* to nuclear *36B4* gene (nDNA) was used to quantify the mtDNA content. The ratio of telomeric repeats copy number (T) to a single copy (36B4) reference gene (S) was used to quantify the telomere length. All the panels show the pre- and post-training measurements for the control group (Con) and for participants with type 1 diabetes (T1D) (female: black circles; male: grey squares). †$P < 0.05$, compared to control.

(Minnock *et al.* 2020). Furthermore, a reduction in the total number of hypoglycaemic events in the post-intervention week suggests that the improved IG control generated by the training positively affects exposure to hypoglycaemia. These results are in line with a recent consensus report which recommends performing COMB exercise in people with type 1 diabetes since this is associated with better glycaemic control (Farinha *et al.* 2017, 2018; Riddell *et al.* 2017). This study showed that the COMB exercise modality applied over a 12-week training period provides a safe and sustainable exercise approach for this population.

Several lines of evidence point to structural, functional and metabolic alterations of the skeletal muscle of both rodent models and humans with type 1 diabetes (Monaco *et al.* 2017), and the molecular data obtained in the present investigation corroborate that work. In our study, participants of the two groups were comparable at baseline having similar age, anthropometric, fitness characteristics, and no evidence of muscular impairment or diabetic complications. However, it was found that mitochondrially encoded *MT-CO2* gene expression, mtDNA/nDNA content and relative protein level of complex I, complex III and complex V of the OXPHOS system were lower in participants with type 1 diabetes. In comparison, Monaco *et al.* (2018) found that participants with type 1 diabetes have a ∼20% reduced

mitochondrial oxidative capacity; however, skeletal muscle mitochondrial size and number, and the levels of individual mitochondrial proteins were not different between people with type 1 diabetes and the control group. The discrepancy between this result and the present study might be explained, at least partially, by the different characteristics of participants. In fact, in the study of Monaco *et al.* participants appeared to be younger, with a shorter duration of diabetes and more physically active (Monaco *et al.* 2018). Although these differences might not be crucial in healthy people, the longer duration of the disease combined with reduced physical activity levels could have accelerated the muscle degeneration and thus manifested a different phenotype (Kalyani *et al.* 2015; Oikawa *et al.* 2019). This observation is supported by the most recent study from Monaco *et al.* that identified impaired OXPHOS oxidative capacity and reduced mitochondrial area density in an older cohort of people with type 1 diabetes (Monaco *et al.* 2021). Furthermore, a recent study from Heyman and colleagues identified a negative relationship between the duration of diabetes, glycaemic control and mitochondrial functioning (Heyman *et al.* 2020). They also identified a blunted oxygen extraction in micro-vessels during exercise, impaired functioning of complex IV of the OXPHOS system and, in agreement with our investigation, suggested the presence of subtle alterations

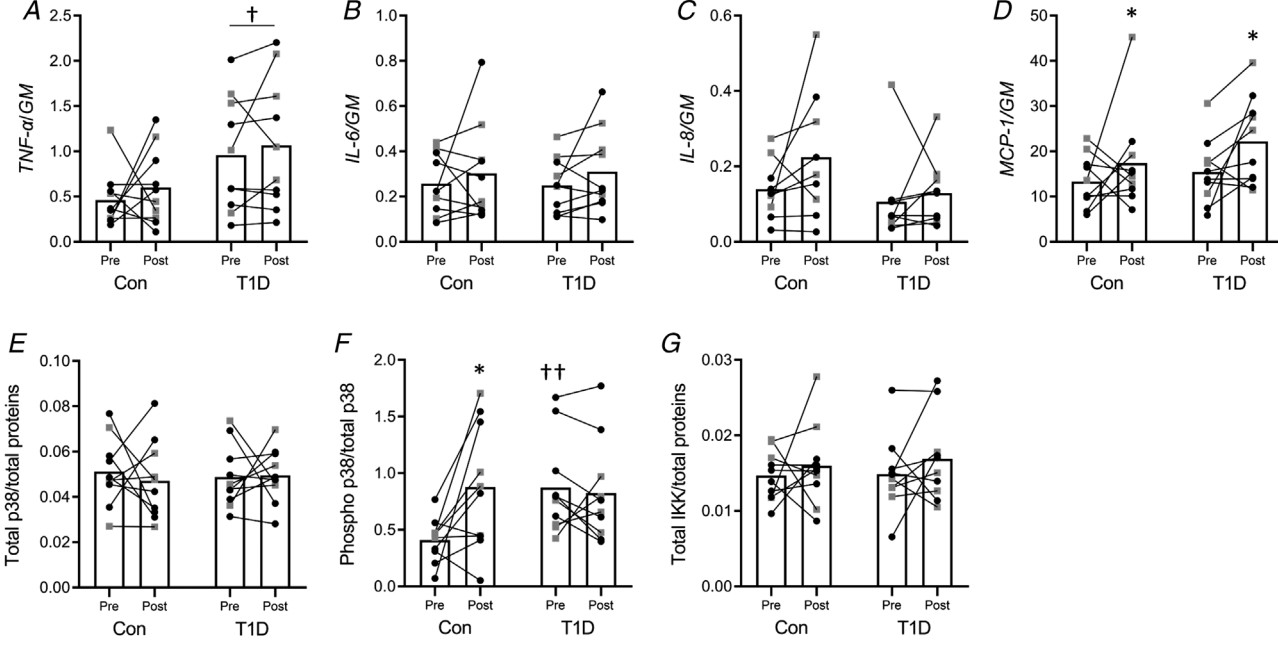

**Figure 4. Markers of inflammation**
*A–D*, muscle mRNA expression levels of *TNF-α* (*A*), *IL-6* (*B*), *IL-8* (*C*) and *MCP-1* (*D*) measured by real-time PCR. The geometric mean (GM) of *GAPDH* and *B2M* mRNA levels was used as the internal reference. *E* and *F*, plot of the densitometric analyses of total (*E*) and phosphorylated (*F*) p38 respectively normalised to total proteins and total p38. *G*, plot of total IKKβ normalised to total proteins. All the panels show the pre- and post-exercise measurement for the control group (Con) and for participants with type 1 diabetes (T1D) (female: black circles; male: grey squares). *$P < 0.05$, compared to Pre; †$P < 0.05$, ††$P < 0.01$, compared to control.

in participants with type 1 diabetes before clinical manifestation.

Furthermore, the present study identified shorter telomere length at baseline in type 1 diabetes compared to the control group, a finding in agreement with the hypothesis of accelerating muscle ageing in this population (Monaco *et al.* 2019). Even though many aspects are involved in cellular senescence, telomere

attrition is a key marker of this process and several studies have already demonstrated the link between oxidative metabolism and telomere length (Kadi & Ponsot, 2010; Oeseburg *et al.* 2010). It is known that exercise might directly modulate telomere length through alteration of redox status and inflammation (Arsenis *et al.* 2017). In this study, the muscle telomere length in type 1 diabetes increased after training but the difference reported was

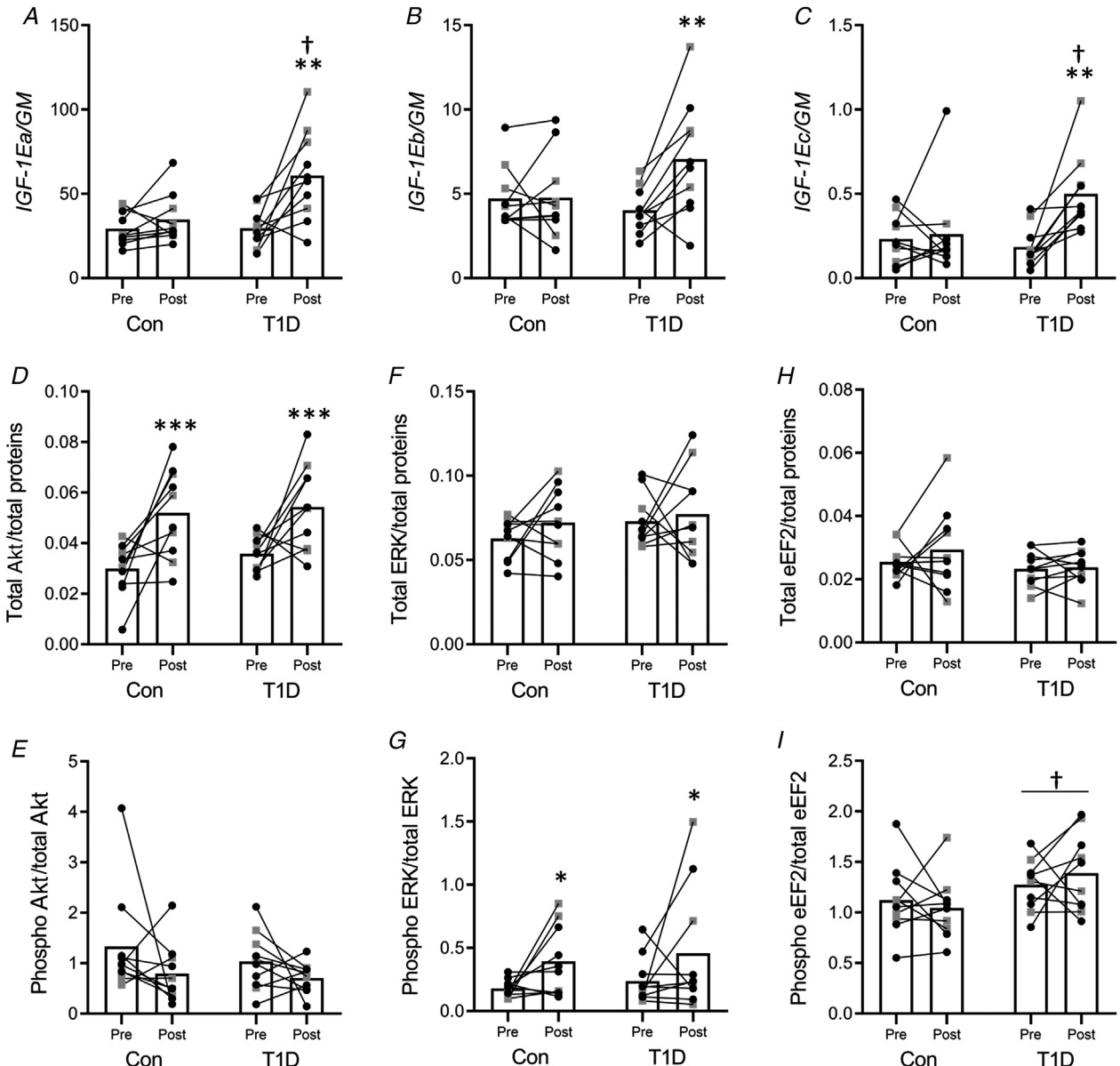

**Figure 5. Markers of muscle growth and remodelling**

*A–C*, muscle mRNA expression levels of the *IGF-1EA* (*A*), *IGF-1Eb* (*B*) and *IGF-1Ec* (*C*) isoforms measured by real-time PCR. The geometric mean (GM) of *GAPDH* and *B2M* mRNA levels was used as the internal reference. *D–I*, densitometric analyses of total Akt (*D*), phosphorylated Akt (*E*), total ERK1/2 (*F*), phosphorylated ERK1/2 (*G*), total eEF2 (*H*) and phosphorylated eEF2 (*I*). Target proteins were normalised to total proteins and the phosphorylated proteins on their corresponding total protein content. All the panels show the pre- and post-training measurements for the control group (Con) and for participants with type 1 diabetes (T1D) (female: black circles; male: grey squares). *$P < 0.05$, **$P < 0.01$, ***$P < 0.001$ compared to Pre; †$P < 0.05$, compared to control.

not significant, probably due to the high interindividual variability.

Physical capacity and exercise training potentially play an important role in the stimulation of mitochondrial biogenesis through effects on *PGC-1a* and AMPK activation (Röckl *et al.* 2008), which in turn improves both glucose and fat oxidation (Reznick & Shulman, 2006). Indeed, we found an increase in AMPK phosphorylation after training in both groups, while other markers of mitochondrial biogenesis and function did not change. The increased AMPK activation is particularly important in type 1 diabetes and suggests that the response to exercise training is partially preserved and might have contributed to the beneficial effects on glucose control observed after training (Cokorinos *et al.* 2017).

On the other hand, the finding that the training programme did not ameliorate the mitochondrial function markers in type 1 diabetes was disappointing. Among various possibilities, we hypothesise that it was due to the relatively short duration (12 weeks) and reduced volume of the training intervention (20 min for AER and RES), as suggested also by insulin dosage that did not change at post-training. In fact, even though the training intervention resulted in a tighter control of glycaemic fluctuations, this was not sufficient to reduce the necessary amount of insulin. However, new studies adopting longer training periods (i.e. 6 months or more) will be necessary to confirm this hypothesis.

Although $\dot{V}_{O_2max}$ improved in both groups, it is important to emphasise that the increase in $\dot{V}_{O_2max}$ was more evident in the control group, suggesting that the reduced mitochondrial functionality found in type 1 diabetes might have contributed to the reduced exercise-induced aerobic adaptations.

Subsequently, as strength adaptation was reduced in type 1 diabetes compared to control, we focused on other important factors associated with muscle quality such as the level of inflammatory markers, muscle growth and regeneration factors. Participants with type 1 diabetes showed higher levels of muscle *TNF-a* mRNA

level and p38 MAPK phosphorylation compared to controls. Inflammation is known to negatively impact skeletal muscle health, as observed by the positive correlation between inflammatory factors and muscle wasting (Ladner *et al.* 2003). Accordingly, an upregulation of the atrophy marker *MuRF-1* mRNA and increased eEF2 phosphorylation, which is associated with reduced translation elongation and protein synthesis (Rose *et al.* 2005), was observed in type 1 diabetes. Moreover, emerging evidence suggests that the state of inflammation in skeletal muscle explains long-term deterioration of muscle mass and function (Schaap *et al.* 2006). Notably, exercise training decreased the *MuRF-1* mRNA quantity, while both *TNF-a* mRNA level and eEF2 phosphorylation (i.e. inactivation) remained higher in type 1 diabetes. *MuRF1* transcription is tightly linked to the development of the atrophy programme and is readily downregulated upon recovery from muscle atrophy (Taillandier & Polge, 2019) and following the expression of anti-inflammatory and anti-catabolic proteins, such as 70 kDa heat shock protein (HSP70) (Farinha *et al.* 2018). Indeed, anabolic effectors like insulin or IGF-1 are able to blunt the expression of *MuRF1* in parallel to muscle sparing during different catabolic situations (Song *et al.* 2019). Here we found an increase of *IGF-1* mRNA isoforms in type 1 diabetes and total Akt and ERK1/2 phosphorylation in both groups after training. The increased *IGF-1* expression and IGF-1 pathway activation observed after COMB exercise training agreed with other studies (Barclay *et al.* 2019; Kraemer *et al.* 2020) and might explain some of the improvements in muscle strength reached after training. The IGF-1 pathways activation might be also related to the partial reduction of inflammation/catabolism observed at post-training in type 1 diabetes (Adams, 2010).

Regarding study limitations, a longer training duration might have produced bigger modifications that were not detectable in this study, and performing the study with more physically active participants might have presented different baseline condition and response to exercise.

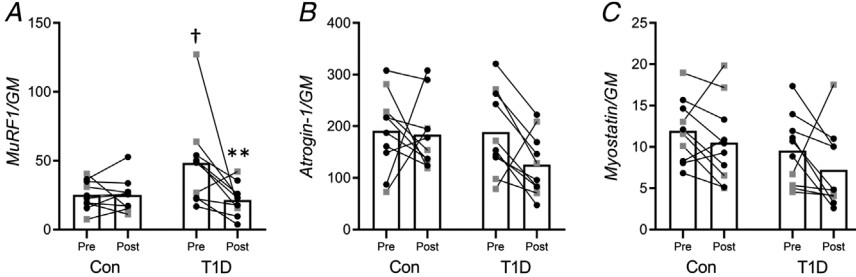

**Figure 6. Markers of muscle atrophy**
Muscle mRNA expression levels of *MuRF1* (*A*), *Atrogin-1* (*B*), and *Myostatin* (*C*). The geometric mean (GM) of *GAPDH* and *B2M* mRNA levels was used as the internal reference. All the panels illustrate the pre- and post-training measurement for the control group (Con) and for participants with type 1 diabetes (T1D) (female: black circles; male: grey squares). \*\**P* < 0.01 compared to Pre; †*P* < 0.05, compared to control.

In conclusion, our findings demonstrated that the COMB exercise modality, applied over a 12-week training period, represents a safe and sustainable exercise approach for people with type 1 diabetes, reducing the risk of glycaemic fluctuations and hypo-glycaemic events. Additionally, we found an alteration of markers of mitochondrial functions, inflammation and growth/atrophy in the muscle of sedentary participants with type 1 diabetes compared to the control group. These molecular differences were accompanied by a reduced exercise adaptation in type 1 diabetes, and we speculate that there might be a relationship between molecular impairments and the exercise response.

The molecular evidence on muscle growth, atrophy and remodelling markers provides novel insights to further our understanding of skeletal muscle biology and health condition in adults with type 1 diabetes. Even though our findings demonstrated the existence of muscle impairment in type 1 diabetes, the positive response to exercise points toward a mild condition and not a severe myopathy.

Emerging studies, together with our evidence, are useful to define the magnitude and the characteristics of diabetic myopathy and can contribute to developing adequate therapeutic strategies that include personalised exercise prescription to maximise muscle health in people with type 1 diabetes.

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

# Additional information

## Data availability statement

All data supporting the results presented in the manuscript are included in the manuscript figures ($n \leq 30$), as per the journal Statistics Policy.

## Competing interests

All authors declare that there are no relationships or activities that might bias, or be perceived to bias, their work.

## Author contributions

D.M. and G.D.V. conceptualised and designed the study, prescribed and supervised the exercise sessions. D.M., G.A., G.V., M.K. and G.D.V. analysed the data and drafted the manuscript. D.M. and G.D.V. collected, processed and stored the biological samples. G.A., R.S., G.V. and E.B. performed the RT-PCR experiments and western blotting. M.K., E.B. and G.D.V. provided overall direction to the project and revised the manuscript. All the authors approved the final version of the manuscript and agree to be accountable for all aspects of the work in ensuring that questions related to the accuracy or integrity of any part of the work are appropriately investigated and resolved. All persons designated as authors qualify for authorship, and all those who qualify for authorship are listed.

## Funding

This research was supported by the Irish Research Council in which D.M. is the recipient of an IRC EBS fellowship award (EBPPG/2016/278). The study sponsor/funder was not involved in the design of the study; the collection, analysis, and interpretation of data; writing the report; and did not impose any restrictions regarding the publication of the report.

## Acknowledgements

Open Access Funding provided by Universita degli Studi di Padova within the CRUI-CARE Agreement.

## Keywords

exercise, exercise physiology, glucose variability, hypoglycaemia, mitochondria, muscle adaptation, myopathy, type 1 diabetes

## Supporting information

Additional supporting information can be found online in the Supporting Information section at the end of the HTML view of the article. Supporting information files available:

**Peer Review History**
**Statistical Summary Document**

