## [Peer Review History · The Journal of Physiology]

Altered muscle mitochondrial, inflammatory and trophic markers and reduced exercise training adaptations in type 1 diabetes

Dean Minnock, Giosuè Annibalini, Giacomo Valli, Roberta Saltarelli, Maurício Krause, Elena Barbieri, and Giuseppe De Vito
DOI: 10.1113/JP282433

Corresponding author(s): Giacomo Valli (giacomo.valli@studenti.unipd.it)

Review Timeline:

Submission Date:	29-Sep-2021
Editorial Decision:	26-Nov-2021
Revision Received:	17-Dec-2021
Accepted:	21-Dec-2021

Senior Editor: Michael Hogan

Reviewing Editor: Bettina Mittendorfer

Transaction Report:

Dear Mr Valli,

Re: JP-RP-2021-282433 "Altered muscle mitochondrial, inflammatory and trophic markers and reduced exercise training adaptations in type 1 diabetes" by Dean Minnock, Giosuè Annibalini, Giacomo Valli, Roberta Saltarelli, Maurício Krause, Elena Barbieri, and Giuseppe De Vito

Thank you for submitting your manuscript to The Journal of Physiology. It has been assessed by a Reviewing Editor and by an expert Referee and I am pleased to tell you that it is considered to be acceptable for publication following satisfactory revision.

The reports are copied at the end of this email. Please address all of the points and incorporate all requested revisions, or explain in your Response to Referees why a change has not been made.

NEW POLICY: In order to improve the transparency of its peer review process The Journal of Physiology publishes online as supporting information the peer review history of all articles accepted for publication. Readers will have access to decision letters, including all Editors' comments and referee reports, for each version of the manuscript and any author responses to peer review comments. Referees can decide whether or not they wish to be named on the peer review history document.

Authors are asked to use The Journal's premium BioRender (<https://biorender.com/>) account to create/redrawn their Abstract Figures. Information on how to access The Journal's premium BioRender account is here: <https://physoc.onlinelibrary.wiley.com/journal/14697793/biorender-access> and authors are expected to use this service. This will enable Authors to download high-resolution versions of their figures.

I hope you will find the comments helpful and have no difficulty returning your revisions within 4 weeks.

Your revised manuscript should be submitted online using the links in Author Tasks Link Not Available.

Any image files uploaded with the previous version are retained on the system. Please ensure you replace or remove all files that have been revised.

REVISION CHECKLIST:

- Article file, including any tables and figure legends, must be in an editable format (eg Word)
- Abstract figure file (see above)
- Statistical Summary Document
- Upload each figure as a separate high quality file
- Upload a full Response to Referees, including a response to any Senior and Reviewing Editor Comments;
- Upload a copy of the manuscript with the changes highlighted.

- A potential 'Cover Art' file for consideration as the Issue's cover image;
- Appropriate Supporting Information (Video, audio or data set https://jp.msubmit.net/cgi-bin/main.plex?form_type=display_requirements#supp).

To create your 'Response to Referees' copy all the reports, including any comments from the Senior and Reviewing Editors, into a Word, or similar, file and respond to each point in colour or CAPITALS and upload this when you submit your revision.

I look forward to receiving your revised submission.

If you have any queries please reply to this email and staff will be happy to assist.

Yours sincerely,

Michael C. Hogan
Senior Editor
The Journal of Physiology
<https://jp.msubmit.net>
<http://jp.physoc.org>
The Physiological Society
Hodgkin Huxley House
30 Farringdon Lane
London, EC1R 3AW
UK
<http://www.physoc.org>
<http://journals.physoc.org>

REQUIRED ITEMS:

-Author photo and profile. First (or joint first) authors are asked to provide a short biography (no more than 100 words for one author or 150 words in total for joint first authors) and a portrait photograph. These should be uploaded and clearly labelled with the revised version of the manuscript. See Information for Authors for further details.

-You must start the Methods section with a paragraph headed Ethical Approval. If experiments were conducted on humans confirmation that informed consent was obtained, preferably in writing, that the studies conformed to the standards set by the latest revision of the Declaration of Helsinki, and that the procedures were approved by a properly constituted ethics committee, which should be named, must be included in the article file. If the research study was registered (clause 35 of the Declaration of Helsinki) the registration database should be indicated, otherwise the lack of registration should be noted as an exception (e.g. The study conformed to the standards set by the Declaration of Helsinki, except for registration in a database.). For further information see: <https://physoc.onlinelibrary.wiley.com/hub/human-experiments>

-Please upload separate high-quality figure files via the submission form.

-You must upload original, uncropped western blot/gel images (including controls) if they are not included in the manuscript. This is to confirm that no inappropriate, unethical or misleading image manipulation has occurred <https://physoc.onlinelibrary.wiley.com/hub/journal-policies#imagmanip> These should be uploaded as 'Supporting information for review process only'. Please label/highlight the original gels so that we can clearly see which sections/lanes have been used in the manuscript figures.

-A Statistical Summary Document, summarising the statistics presented in the manuscript, is required upon revision. It must be on the Journal's template, which can be downloaded from the link in the Statistical Summary Document section here: https://jp.msubmit.net/cgi-bin/main.plex?form_type=display_requirements#statistics

-Please include an Abstract Figure. The Abstract Figure is a piece of artwork designed to give readers an immediate understanding of the research and should summarise the main conclusions. If possible, the image should be easily 'readable' from left to right or top to bottom. It should show the physiological relevance of the manuscript so readers can assess the importance and content of its findings. Abstract Figures should not merely recapitulate other figures in the manuscript. Please try to keep the diagram as simple as possible and without superfluous information that may distract from the main conclusion(s). Abstract Figures must be provided by authors no later than the revised manuscript stage and should be uploaded as a separate file during online submission labelled as File Type 'Abstract Figure'. Please ensure that you include the figure legend in the main article file. All Abstract Figures should be created using BioRender. Authors should use The Journal's premium BioRender account to export high-resolution images. Details on how to use and access the premium account are included as part of this email.

EDITOR COMMENTS

Reviewing Editor:

The reviewer and I myself found considerable merit in the work

Enclosed are specific comments to help further improve this already strong paper

REFEREE COMMENTS

Referee #1:

In the present study, those with and without type 1 diabetes were evaluated before and after 12 weeks of COMB exercise training. Measures were IG variability, hypoglycaemic event number and gene expression of a number of muscle specific signalling pathways, as well as Western blotting for OXPHOS and AMPK.

The inclusion of muscle biopsies pre and post training provide an novel insight into potential changes that are happening in those with T1D. The glycemic data, while a focus of this work, is less novel but does serve to demonstrate an important patient focused outcome.

Major Comments:

(1) Authors briefly discuss the low N for males and females prohibited a more detailed investigation. While this may be true, the results of the present study would be more important and novel for the research community (and the lay reader) if data points were coded to denote female and male participants. This would allow the reader to observe sex differences between those with type 1 diabetes and those without.

(2) The primary outcome of interest of this work was noted as the changes in glycemia. This outcome has been studied extensively (and, as the authors noted, has a consensus report written on the subject). The novel aspect of this work (muscle changes) should be the primary outcome of interest with the changes in glycemic a secondary outcome. This would also make the introduction more relevant (which focuses primarily on skeletal muscle)

(3) Given the recent work of Dial, Grafham et al (2021- AJP Cell), do you think that the reduced benefits of RE in those with type 1 diabetes was related to a reduced ability to repair from the previous bouts of exercise?

Minor Comments;

- [] Might be more impactful if the subject dots (pre and post) were connected together to show the change for each subject.
- [] It is admittedly surprising to see significant changes in so many measures of aerobic and resistance exercise fitness yet no change in measures that subjects would care about (LBM, BMI, BW). Why do the authors think this happened?
- [] L109: 'muscle to' not 'to muscle'
- [] was there a significant differential between people (in METs) before and during the exercise training sessions. Essentially, subjects were doing less than 120 min and <500METs before. Now they are doing 120 min and 500 METs. Not a huge difference possibly? Also, why did you choose training only 3 times a week as this would not reach the recommendations of most diabetes organizations worldwide?
- [] Figure 3B- Y axis has a spelling mistake (length, not lenght)
- [] Increase the size of the asterisks on graphs to distinguish from data points.
- [] any figure legends include statistical descriptors which are not found in the data. These should be omitted.
- [] L422-423: "... (Sivitz & Yorek, 2010) and the molecular data obtained in the present investigation corroborate these works." This is not the appropriate reference for what the authors are proposing. In particular because it focuses on mitochondria and because it is too dated to represent the studies completed in humans in recent years.
- [] L440: Monaco et al 2021 is the appropriate reference for this statement
- [] L443: should not be the beginning of a new paragraph

END OF COMMENTS

Confidential Review

29-Sep-2021

Dear editor and reviewers, we the authors very much appreciated the opportunity to publish in *The Journal of Physiology* and are grateful for the suggestions to improve our paper. Whether you might not be satisfied with any of the answers or changes provided, we remain available for further clarification.

Please see our reply below:

REQUIRED ITEMS:

-Author photo and profile. First (or joint first) authors are asked to provide a short biography (no more than 100 words for one author or 150 words in total for joint first authors) and a portrait photograph. These should be uploaded and clearly labelled with the revised version of the manuscript. See Information for Authors for further details.

The item was uploaded

-You must start the Methods section with a paragraph headed Ethical Approval. If experiments were conducted on humans confirmation that informed consent was obtained, preferably in writing, that the studies conformed to the standards set by the latest revision of the Declaration of Helsinki, and that the procedures were approved by a properly constituted ethics committee, which should be named, must be included in the article file. If the research study was registered (clause 35 of the Declaration of Helsinki) the registration database should be indicated, otherwise the lack of registration should be noted as an exception (e.g. The study conformed to the standards set by the Declaration of Helsinki, except for registration in a database.). For further information see:

<https://physoc.onlinelibrary.wiley.com/hub/human-experiments>

This point was changed in the text

-Please upload separate high-quality figure files via the submission form.

The item was uploaded

-You must upload original, uncropped western blot/gel images (including controls) if they are not included in the manuscript. This is to confirm that no inappropriate, unethical or misleading image manipulation has occurred <https://physoc.onlinelibrary.wiley.com/hub/journal-policies#imagmanip> These should be uploaded as 'Supporting information for review process only'. Please label/highlight the original gels so that we can clearly see which sections/lanes have been used in the manuscript figures.

The item was uploaded

-A Statistical Summary Document, summarising the statistics presented in the manuscript, is required upon revision. It must be on the Journal's template, which can be downloaded from the link in the Statistical Summary Document section here: https://jp.msubmit.net/cgi-bin/main.plex?form_type=display_requirements#statistics

The item was uploaded

For the sake of clarity, we changed some statistical symbols making them homogeneous across all tables and figures as follow:

* $p < 0.05$, ** $p < 0.01$, *** $p < 0.001$ compared to pre;

† $p < 0.05$, †† $p < 0.01$, ††† $p < 0.001$, compared to control.

This should make the identification of the statistical differences easier.

The reviewer and the editor should note that during the filling of the statistical summary document and the careful revision of the manuscript, we realised 2 mistakes:

1. we have incorrectly reported MCP-1 and IGF-1 mRNA expression (in the Methods, Statistical Analysis section) as not normally distributed variables. We corrected this error in the revised manuscript.
2. We noticed that carbohydrates intake was higher in type 1 diabetes compared to the control group (but no difference between pre and post-training was observed) and this was not reported in the manuscript. This is now reported in the results section.

Please note that none of these changes modified the results described in the first version of the manuscript.

We remain available for any additional clarification.

-Please include an Abstract Figure. The Abstract Figure is a piece of artwork designed to give readers an immediate understanding of the research and should summarise the main conclusions. If possible, the image should be easily 'readable' from left to right or top to bottom. It should show the physiological relevance of the manuscript so readers can assess the importance and content of its findings. Abstract Figures should not merely recapitulate other figures in the manuscript. Please try to keep the diagram as simple as possible and without superfluous information that may distract from the main conclusion(s). Abstract Figures must be provided by authors no later than the revised manuscript stage and should be uploaded as a separate file during online submission labelled as File Type 'Abstract Figure'. Please ensure that you include the figure legend in the main article file. All Abstract Figures should be created using BioRender. Authors should use The Journal's premium BioRender account to export high-resolution images. Details on how to use and access the premium account are included as part of this email.

The item was uploaded

EDITOR COMMENTS

Reviewing Editor:

The reviewer and I myself found considerable merit in the work

Enclosed are specific comments to help further improve this already strong paper

REFEREE COMMENTS

Referee #1:

In the present study, those with and without type 1 diabetes were evaluated before and after 12 weeks of COMB exercise training. Measures were IG variability, hypoglycaemic event number and gene expression of a number of muscle specific signalling pathways, as well as Western blotting for OXPHOS and AMPK.

The inclusion of muscle biopsies pre and post training provide an novel insight into potential changes that are happening in those with T1D. The glycemic data, while a focus of this work, is less novel but does serve to demonstrate an important patient focused outcome.

Major Comments:

(1) Authors briefly discuss the low N for males and females prohibited a more detailed investigation. While this may be true, the results of the present study would be more important and novel for the research community (and the lay reader) if data points were coded to denote female and male participants. This would allow the reader to observe sex differences between those with type 1 diabetes and those without.

We agree with this comment as the gender-specific response is an important topic. At first, we decided not to distinguish between males and females because of the small sample size that prevented the application of appropriate statistical analysis on the two separated groups. However, in response to this comment, male and female participants were separated in the figures (different colours and shapes) in order to allow the reader to clearly distinguish between the two.

(2) The primary outcome of interest of this work was noted as the changes in glycemia. This outcome has been studied extensively (and, as the authors noted, has a consensus report written on the subject). The novel aspect of this work (muscle changes) should be the primary outcome of interest with the changes in glycemic a secondary outcome. This would also make the introduction more relevant (which focuses primarily on skeletal muscle)

We really agree with this comment. It was changed in the text

(3) Given the recent work of Dial, Grafham et al (2021- AJP Cell), do you think that the reduced benefits of RE in those with type 1 diabetes was related to a reduced ability to repair from the previous bouts of exercise?

This is a very good point. In our previous study (Minnock et al., 2020; Eur J Appl Physiol. 2020 Dec;120(12):2677-2691), we demonstrated that an acute bout of combined resistance and aerobic exercise (the same as used in the present study) did not increase muscle markers of muscle damage (serum creatine kinase and LDH) in type 1D subjects. Thus, the exercise modality adopted in our study differed completely from that reported in Dial et al (2021- AJP Cell), which used an exercise protocol specifically designed to induce muscle damage (eccentric quadriceps contractions). In fact, Dial et al. observed a large increase 96 hours post-exercise of serum CK (75 times higher than resting levels). The exercise protocol proposed in our study was well tolerated by all participants and no injuries events were recorded during the training period. Moreover, only minor differences were observed for the session RPE assessed across the 12 weeks of training. Obviously, the experimental design of the present study, which considers only pre and post-training muscle biopsies, did not allow us to draw conclusive remarks on this issue. Further studies, which may consider multiple biopsies during the training sessions, might be useful to clarify this point. If the reviewer and the editor believe that this aspect should be added in the discussion section, we can add a sentence on that.

Minor Comments;

- [] Might be more impactful if the subject dots (pre and post) were connected together to show the change for each subject.

We consider this a good suggestion, the figures were changed

- [] It is admittedly surprising to see significant changes in so many measures of aerobic and resistance exercise fitness yet no change in measures that subjects would care about (LBM, BMI, BW). Why do the authors think this happened?

No physical characteristic differences were found between type 1 diabetes and control participants at baseline. Moreover, no significant improvements in lean mass, and body fat, were observed following the intervention period. This, in part, can be explained by the brief duration of the exercise sessions (40mins),

resulting in an overall low volume of total exercise performed over 36 exercise sessions. It should be noted that we monitored carbohydrate intake in both groups at Pre- and Post-intervention week (as it might heavily affect glucose control) but did not control for energy intake across all the study duration. This could have also affected body composition adaptations. These findings, however, confirm previous research, indicating that HIT exercise training may not be the most effective modality towards improving lean body mass, body fat (kg) and body fat percentage (Sultana et al., 2019), mostly due to their short duration and overall shortened volume of exercise when compared to traditional training methods.

- [] L109: 'muscle to' not 'to muscle'

This was corrected

- [] was there a significant differential between people (in METs) before and during the exercise training sessions. Essentially, subjects were doing less than 120 min and <500METs before. Now they are doing 120 min and 500 METs. Not a huge difference possibly? Also, why did you choose training only 3 times a week as this would not reach the recommendations of most diabetes organizations worldwide?

The primary reason for training 3 times weekly was for the convenience of the participants. Our aim was to mimic a real-world scenario and our belief was that by offering a manageable and realistic protocol our participants who were not habitual exercisers would adhere to the programme. Moreover, the aim was not to reach the desired recommended training durations set out by diabetes organisations. Based on our previous research (Minnock et al., 2020) which indicated that 40 min combined exercise resulted in the safest glycaemic response in the 24 hour post exercise. We wished examine the physical and molecular outcomes of this training mode when applied to a chronic training intervention.

- [] Figure 3B- Y axis has a spelling mistake (length, not lenght)

This was corrected

- [] Increase the size of the asterisks on graphs to distinguish from data points.

This was changed

For the sake of clarity, we changed some statistical symbols making them homogeneous across all tables and figures as follow:

* $p < 0.05$, ** $p < 0.01$, *** $p < 0.001$ compared to pre;

† $p < 0.05$, †† $p < 0.01$, ††† $p < 0.001$, compared to control.

This should make the identification of the statistical differences easier.

- [] any figure legends include statistical descriptors which are not found in the data. These should be omitted.

This was corrected

- [] L422-423: "... (Sivitz & Yorek, 2010) and the molecular data obtained in the present investigation corroborate these works." This is not the appropriate reference for what the authors are proposing. In particular because it focuses on mitochondria and because it is too dated to represent the studies completed in humans in recent years.

This was corrected

- [] L440: Monaco et al 2021 is the appropriate reference for this statement

This was corrected

- [] L443: should not be the beginning of a new paragraph

This was corrected

Dear Dr Valli,

Re: JP-RP-2021-282433R1 "Altered muscle mitochondrial, inflammatory and trophic markers and reduced exercise training adaptations in type 1 diabetes" by Dean Minnock, Giosuè Annibalini, Giacomo Valli, Roberta Saltarelli, Maurício Krause, Elena Barbieri, and Giuseppe De Vito

I am pleased to tell you that your paper has been accepted for publication in The Journal of Physiology.

NEW POLICY: In order to improve the transparency of its peer review process The Journal of Physiology publishes online as supporting information the peer review history of all articles accepted for publication. Readers will have access to decision letters, including all Editors' comments and referee reports, for each version of the manuscript and any author responses to peer review comments. Referees can decide whether or not they wish to be named on the peer review history document.

Are you on Twitter? Once your paper is online, why not share your achievement with your followers. Please tag The Journal (@jphysiol) in any tweets and we will share your accepted paper with our 23,000+ followers!

The last Word version of the paper submitted will be used by the Production Editors to prepare your proof. When this is ready you will receive an email containing a link to Wiley's Online Proofing System. The proof should be checked and corrected as quickly as possible.

Authors should note that it is too late at this point to offer corrections prior to proofing. The accepted version will be published online, ahead of the copy edited and typeset version being made available. Major corrections at proof stage, such as changes to figures, will be referred to the Reviewing Editor for approval before they can be incorporated. Only minor changes, such as to style and consistency, should be made a proof stage. Changes that need to be made after proof stage will usually require a formal correction notice.

All queries at proof stage should be sent to TJP@wiley.com

Yours sincerely,

Michael C. Hogan
Senior Editor
The Journal of Physiology
<https://jp.msubmit.net>
<http://jp.physoc.org>
The Physiological Society
Hodgkin Huxley House
30 Farringdon Lane
London, EC1R 3AW
UK
<http://www.physoc.org>
<http://journals.physoc.org>

P.S. - You can help your research get the attention it deserves! Check out Wiley's free Promotion Guide for best-practice recommendations for promoting your work at www.wileyauthors.com/eeo/guide. And learn more about Wiley Editing Services which offers professional video, design, and writing services to create shareable video abstracts, infographics, conference posters, lay summaries, and research news stories for your research at www.wileyauthors.com/eeo/promotion.

* IMPORTANT NOTICE ABOUT OPEN ACCESS *

Information about Open Access policies can be found here <https://physoc.onlinelibrary.wiley.com/hub/access-policies>

To assist authors whose funding agencies mandate public access to published research findings sooner than 12 months after publication The Journal of Physiology allows authors to pay an open access (OA) fee to have their papers made freely available immediately on publication.

You will receive an email from Wiley with details on how to register or log-in to Wiley Authors Services where you will be able to place an OnlineOpen order.

You can check if your funder or institution has a Wiley Open Access Account here <https://authorservices.wiley.com/author-resources/Journal-Authors/licensing-and-open-access/open-access/author-compliance-tool.html>

Your article will be made Open Access upon publication, or as soon as payment is received.

If you wish to put your paper on an OA website such as PMC or UKPMC or your institutional repository within 12 months of publication you must pay the open access fee, which covers the cost of publication.

OnlineOpen articles are deposited in PubMed Central (PMC) and PMC mirror sites. Authors of OnlineOpen articles are permitted to post the final, published PDF of their article on a website, institutional repository, or other free public server, immediately on publication.

Note to NIH-funded authors: The Journal of Physiology is published on PMC 12 months after publication, NIH-funded authors DO NOT NEED to pay to publish and DO NOT NEED to post their accepted papers on PMC.

PEER REVIEW COORDINATOR COMMENTS:

Note: at the publication proofing stage, please could authors add a statement on database registration and compliance with Clause 35 (of the Helsinki Declaration), or else state "except for registration in a database."

EDITOR COMMENTS

Reviewing Editor:

No further comments

REFEREE COMMENTS

Referee #1:

The authors were very attentive to the recommendations suggested. No further comments.

This research will be a welcome addition to the muscle research field in those with Type 1 Diabetes.

END OF COMMENTS

1st Confidential Review

17-Dec-2021